# How Should Job Crafting Interventions Be Implemented to Make Their Effects Last? Protocol for a Group Concept Mapping Study

**DOI:** 10.3390/ijerph192113922

**Published:** 2022-10-26

**Authors:** Marta Roczniewska, Emma Hedberg Rundgren, Henna Hasson, Arnold B. Bakker, Ulrica von Thiele Schwarz

**Affiliations:** 1Medical Mangement Centre, Department of Learning, Informatics, Management, and Ethics, Karolinska Institutet, 171 77 Stockholm, Sweden; 2Institute of Psychology, SWPS University of Social Sciences and Humanities, 03-815 Warsaw, Poland; 3Centre for Epidemiology and Community Medicine, 104 31 Stockholm, Sweden; 4Center of Excellence for Positive Organizational Psychology, Erasmus University Rotterdam, 3000 DR Rotterdam, The Netherlands; 5Department of Industrial Psychology and People Management, University of Johannesburg, Johannesburg 2006, South Africa; 6School of Health, Care and Social Welfare, Mälardalen University, 721 23 Västerås, Sweden

**Keywords:** job crafting, interventions, implementation, group concept mapping

## Abstract

Background: By means of job crafting (JC) employees shape and customize their job design to align it with their preferences. Research has so far shown that such bottom-up proactivity can be stimulated via JC interventions. While the overall effectiveness behind these interventions has been supported, it is unclear how to implement these interventions to make their effects lasting. Methods: The overall aim of this project will be to investigate how to implement JC interventions with lasting effects. We will apply a group concept mapping (GCM) methodology, which is a mixed methods approach of exploratory nature for engaging stakeholder groups in a structured conceptualization process. As part of concept mapping procedures, brainstorming sessions will be conducted with experts in job crafting to identify factors expected to make job crafting intervention effects lasting. These factors will be sorted by similarity and rated by each participant in regard to their perceived importance and feasibility to ensure lasting, sustainable effects. The data will be analyzed using multidimensional scaling (MDS), hierarchical cluster analysis, and descriptive and inferential statistics, resulting in a visual representation of conceptually distinguished clusters representing the factors influencing the sustainability of JC interventions. In the final step, a workshop will be conducted with the participants to facilitate the interpretation of the results. Results and conclusion: This study will provide knowledge relevant to organizational practitioners and scholars who want to implement JC interventions with lasting effects. Although data collected following the group concept mapping procedure is self-reported and at risk of being simplified, the method allows for a structured conceptualization process integrating different perspectives and uncovering implicit knowledge making it suitable for studying complex phenomena. The results will not only enrich the current literature concerning the effectiveness of JC interventions but also be used to develop a practitioner-oriented toolkit outlining evidence-based recommendations concerning designing and implementing, as well as evaluating JC interventions.

## 1. Introduction

The term ‘job design’ denotes the way in which ‘jobs, tasks, and roles are structured, enacted, and modified’ [1] (p. 319), while ‘redesign’ is the process through which some aspects of the job, the tasks or the roles are modified [2]. Traditionally, the job redesign process has been perceived as a top-down process, i.e., where jobs are redesigned by the organization for the employee [3] Yet, such attempts suffer from several drawbacks. First, the changing nature of jobs and the pace of these changes make the top-down process inadequate for timely reactions. Second, the top-down approach neglects the growing specialization of jobs and the complexity of organizational processes, which may be the reason why strict ‘traditional’ job redesign frameworks showed mixed effectiveness [4,5]. Third, the workforce is diverse and a ‘one size fits all’ approach seems unsuitable to redesign the jobs in a way that makes them engaging and meaningful for different individual employees given their diverse needs or preferences. 

These deficiencies have sparked interest in more individualized, bottom-up job redesign approaches that recognize the role of individual employees as proactive agents who shape their jobs. One such approach is job crafting (JC)*:* proactive behavior that enables individuals to change their work characteristics to fit them with their needs and preferences [2,6], as well as to adapt to organizational changes [7]. Wrzesniewski and Dutton [6], who coined this concept, define JC as proactive modifications in physical, relational, and cognitive aspects of one’s job. Employees who craft their jobs may alter the boundaries of their tasks, modify the nature and intensity of social interactions, or change how they perceive and understand their role at work. The second approach frames JC within Job Demands-Resources (JD-R) theory [8,9,10] and defines it as behaviors aimed at changing two types of job characteristics: job demands and job resources to find a better person-job fit [2]. Specifically, employees may seek more structural and social job resources, increase challenging job demands, and reduce or optimize hindering job demands [11,12]. 

Further conceptualizations of JC have distinguished between redesign attempts aimed at reaching perceived gains and increasing certain job characteristics (promotion crafting), versus that concentrating on avoiding perceived negative end-states and reducing unwanted job aspects (prevention crafting) [13]. A recent attempt to synthesize these distinct streams of research on JC has proposed a three-level hierarchical structure of this construct [14]. The first level describes JC orientation: approach (enriching and expanding) versus avoidance (reducing and limiting). The second designates JC form, i.e., behavioral (changes in actions) versus cognitive (changes in perceptions). The third level concerns JC content: altering the levels of job resources or job demands. Thus, for example, when employees seek more social job resources, it represents an approach-, behavioral-, and resources-focused strategy.

Meta-analyses have linked approach JC with positive outcomes, such as job satisfaction, work engagement, and performance [13,15], while avoidance of JC has been linked with negative outcomes, such as higher strain and turnover intentions [13,15] Those who craft their jobs experience stronger person-job fit [16] and improved meaning of work [17]. Promotion-focused JC has been linked with better health [18] and lower absenteeism [19]. JC has been particularly effective in buffering the negative effects of job demands on burnout [20]. Research also shows that employees can craft their way through organizational change to better adjust to the new reality [7,21]. This role of JC has been observed during the recent COVID-19 pandemic, where—forced to telework—many employees had to redesign their jobs to match the new circumstances, deal with the new demands, and achieve work-life balance [22]. As a result, JC may be linked with more sustainable employment [23,24]. These promising findings have encouraged researchers and practitioners alike to investigate ways in which JC efforts can be fostered.

Earlier research has shown that employees can learn how to craft their jobs in a training situation and then transfer these behaviors to work settings. Such JC interventions are training interventions and have most frequently been conducted in group workshop settings [25,26,27,28], and more recently also as e-interventions [29]. Additionally, there is a ‘Job Crafting Exercise’ [30]—a self-developmental paper-and-pencil tool that helps individuals recognize ways in which they could craft their jobs to better fit their motives, strengths, and passions. Typically, JC interventions involve several steps. Many versions of the JC intervention contain a job and/or person analysis, where employees are asked to reflect on their values, their tasks and responsibilities, available organizational, job, and personal resources (e.g., strengths or passions), and their job demands. Often, theories on job crafting are explained and participants share examples of their earlier crafting attempts to inspire and learn from one another. JC interventions usually comprise an action plan where individuals are asked to focus on their own job and identify tasks or work characteristics (demands or resources) that they would like to change using crafting in a specific period of time. Some interventions use homework assignments or prompts (nudges) to keep employees engaged in JC after the workshop sessions [29,31,32]. Many interventions comprise a reflection phase where after weeks or months of practicing job crafting in real work situations, employees reconvene to evaluate these attempts, receive feedback from others, and make new JC plans including identified barriers.

A recent meta-analysis with utility analysis of 14 group JC interventions has demonstrated that participation in a JC intervention (compared to a control condition) is linked with more frequent JC attempts for seeking challenges and reducing demands. The findings have additionally shown that JC interventions could be considered effective in helping employees to achieve better work engagement and be more proactive (i.e., higher contextual performance), but did not boost task or adaptive performance [33].

One major limitation of the aforementioned meta-analysis [33] is that majority of the included studies included relatively short-term follow-ups. As a result, we know little about the long-term effectiveness of the JC interventions: when (if at all) does the effect disappear? Are any versions of the JC interventions linked with more sustainable outcomes? Which steps or activities are crucial? Do JC interventions fit with some contexts more than others? Are JC interventions more effective for particular groups of employees? For instance, the meta-analysis found that interventions in which participants formed plans that included both organizational and personal objectives (vs. solely personal) had moderate effectiveness in boosting work engagement. It also found that JC interventions were more effective in improving task performance among healthcare professionals than educators. These preliminary findings show that there could be potential context, employee, intervention, and implementation factors to consider when one introduces JC interventions.

While guidelines regarding implementing, designing, and evaluation exist for organizational interventions [34], we know little about the implementation of JC interventions specifically. This knowledge gap is relevant because, as we believe, sustainable JC interventions require two-way processes. One is a bottom-up perspective: the continuous process in which employees engage in job redesign to increase their job fit and satisfaction. The other is a top-down perspective: the modes by which organizations can prepare the environment to enable employees to craft their jobs after “they are back to work” from the intervention. How should these two perspectives be aligned to work towards a common cause? Given that JC behaviors or strategies differ between individuals and may go unnoticed by managers [6], identifying effective supporting organizational interventions is a challenge. To address these knowledge gaps, we will perform an exploratory study among practitioners and researchers who perform JC interventions to identify factors that they find relevant for the implementation of JC interventions from the perspective of their sustainability.

The following research questions (RQ) will be investigated:

RQ1. What factors make JC interventions have lasting effects?

RQ2. What is the perceived importance of each of these factors?

RQ3. What is the perceived feasibility of ensuring each of these factors?

## 2. Methods

To answer the research questions Group Concept Mapping, GCM, will be used [35]. GCM is “a structured process, focused on a topic or construct of interest, involving input from multiple participants, that produces an interpretable pictorial view of their ideas and concepts and how these are interrelated” [36]. GCM is a generic integrative mixed method for conceptualization, resulting in a visual representation of a conceptual framework. The methodology builds on Trochim and Linton’s [37] general model of conceptualization and engages the participants in an iterative conceptualization process in which the participants take part in both data collection, analysis, and interpretation of the results [35]. Thus, by definition, structurization and multiple perspectives are essential parts of this process. Structurization results in the concept mapping process being transparent, but also allows for managing a group process that involves multiple participants. This mixed-method approach is well-established in different disciplines and settings. Because of its generic nature, the methodology is suitable for understanding a variety of areas or topics and has previously been used across research areas including, i.e., health care, psychology, implementation, and social welfare [38,39,40].

Through GCM, experts can explore, structure, and prioritize the gained data with their own perspective. They also receive an overall ‘picture’ of the topic. The method at hand can be seen as a learning strategy to promote collaborative learning [41]. In the manner of other mixed methods, the qualitative and quantitative data complement each other [42].

GCM is a mixed method involving both qualitative and quantitative techniques resulting in a nuanced understanding of the studied phenomena. The GCM procedure is suitable for studying complex phenomena that call for the incorporation of different perspectives, i.e., both those of practitioners and academics [35]. GCM was chosen here because it allows us to display the knowledge of experts from different backgrounds and gain a broad picture of their assessment of important influencing factors for JC interventions.

Appendix A contains a checklist of reporting guidelines for mixed-method research (supplemented with specific items for concept mapping) that we completed for the study.

### 2.1. Participants and Procedure

To identify conditions that ensure the sustained impact of job crafting interventions, experts in job crafting will be invited to participate in the group concept mapping procedure. The experts will be identified based on (1) authorship of research articles about JC interventions or, (2) practical experience with job redesign interventions. We will employ three recruitment methods. First, we have already identified empirical articles demonstrating results from job crafting interventions. We will contact the first authors as their e-mails are provided. Second, these authors will also be asked to forward the study link to other researchers that they see suitable. Third, as job rafting researchers ourselves, we are aware of other researchers in the field, and we will contact them directly to invite them to participate. If the participants have either published a research article about JC interventions or have practical experience with organizational interventions no exclusion criteria will be used. The practitioners will be identified by contacting researchers who have used their help in implementing JC interventions, as well as through a network of JC facilitators trained by a consultancy employing work psychologist studying and working with JC (https://job-crafting.pl/). The GCM-procedure includes three data collection activities and a workshop which aims to help facilitate the interpretation of the results. The participants will be invited to participate in each step of the procedure. However, it is not necessary for all participants to participate in all the steps of the process. Although there is no strict limit on how many participants should take part in each data collection activity to generate meaningful results, a minimum of at least 10 participants from each stakeholder group is recommended to be included in each data collection activity [35]. In this study, we aimed to include 40 researchers and 40 organizational practitioners in each data collection activity.

The GCM methodology includes three data collection activities: (1) Brainstorming, (2) Sorting, and (3) Rating. The data collection activities are centered around a pre-decided focus prompt that guides the data collection activities, e.g., “What factors are needed for job crafting interventions to have lasting effects?”. This preliminary focus prompt will be revised by the research team who are experienced with both jobs crafting interventions and the GCM procedure based on pilot-testing among representatives of the target group to ensure it is clear and adheres to the focus of the project, and allows for divergent thinking. For matters of feasibility, the GCM will be conducted via the online tool. We will use the Concept Systems groupwisdom™ web platform to undertake the GCM steps. All data collection activities will be conducted anonymously.

The first data collection activity in the GCM methodology is the brainstorming activity. The task in the brainstorming activity will be to generate short statements that describe what is needed to make job crafting interventions sustainable (e.g., Managers should support crafting attempts; JC should solve actual organizational problems; Employees’ skills discretion should be increased). Each participant will generate statements by answering the focus prompt individually and is instructed to generate as many meaningful statements as possible. When all participants have generated statements, the research team will compile a comprehensive list of all generated statements by breaking up statements that contain more than one element and removing duplicates, statements that are highly overlapping, and statements that are incomprehensible.

The second data collection activity is the sorting activity. In the sorting activity, the participants will be asked to individually sort all of the statements included in the comprehensive list into piles based on similarity in a way meaningful to the participants, yielding as many piles as each participant deem appropriate.

In the last data collection activity, the rating activity, participants will be asked to rate each statement on a 5-point Likert scale in regard to how important, as well as how feasible, each guideline/suggestion would be to ensure when implementing job crafting interventions. “Importance” aims to obtain a response regarding the value, worth, and consequence of the issue, whereas “feasibility” concerns the extent to which the issue is possible to address in practice. To minimize the workload on the participants the rating activity will be conducted approximately one month after the sorting activity.

### 2.2. Data Management

All data will be collected and analyzed using the software groupwisdom (Concept systems incorporated, 2022). Access to the software will be password protected and the data will be accessible only to the research group through personal passwords. Final storage of the data collected will be made using an electronic notebook for research documentation for long-term retention. There is no upper time limit for data storage.

### 2.3. Analytical Strategy

The sorting data will be analyzed using multidimensional scaling (MDS) and hierarchical cluster analysis to create a visual representation of how the statements relate to each other [36]. Data from the sorting activity will be used to create an overall square symmetric similarity matrix in which the value of each cell represents how many times each pair of statements has been sorted together by the participants. To make the similarity matrix more interpretable, MDS analysis will be conducted to create a two-dimensional representation, a point map, of the similarity matrix. On the point map, each statement is represented by a point on a two-dimensional graph. The placement on each point is based on how many times the statement has been sorted with all other statements. An example of a point map and a cluster solution is presented in Figure 1.

To determine how well the two-dimensional representation fits the original data, a *stress value* will be calculated. The stress value is an indicator of how much disparity it is between the two-dimensional point map and the similarity matrix, with lower values indicating a better fit [35]. Both the number of participants and the consensus amongst them influence the stress value. If the consensus is high, the stress value is low and the more participants included in the analysis, the lower the stress value. Stress values between 0.10 and 0.35 are considered to generate meaningfully interpretable results [35].

While the stress value indicates the overall fit of all statements on the point map in relation to the original data, there is also variation in how well each point fits the overall similarity matrix. To measure how well the position of each point fits the original data, the *bridging value* will be calculated for each point [35]. Statements with low bridging values will have been sorted more similarly by the participants than statements with high bridging values. High bridging values indicate the statement is possibly ambiguous or related to several different concepts within the studied phenomena.

When the point map has been created, the X- and the Y-coordinate for each point will be used as input for an agglomerative cluster analysis using the Wards algorithm [35]. It is possible to create as many clusters as statements included in the analysis. By using the algorithm, clusters will be merged by minimizing the sum of squares of the distances between all statements in any two hypothetical clusters that might be merged, when going from a higher to a lower cluster solution, i.e., when going from a 10-cluster solution to a 9-cluster solution. The process will start at the maximum cluster solution and merge clusters one by one. The research group will evaluate the cluster solutions by investigating at what cluster level creating an additional cluster will not increase the meaningfulness of the cluster map and merging two clusters will decrease the meaningfulness of the cluster map.

Further, the rating results will be used to create a go-zone display [35]. The go-zone display is an X-Y graph that compares ratings. The bivariate graph shows the average rating of each statement on the two variables included in the analysis: importance and feasibility. The graph is divided into quadrants by the mean rating value of each variable. The upper right quadrant will include statements that are rated above average on both importance and feasibility, meaning this zone contains statements of special interest. This zone is called the go-zone. The go-zone display can be used to identify statements of interest when using the results for planning and evaluation. Further, we will use inferential statistics to investigate differences in researchers’ and practitioners’ perceptions of the importance and feasibility of the statements/clusters derived from the analysis.

The participants will be invited to a digital workshop to help facilitate the interpretation of the cluster solution. The cluster solution will be presented to the participants who will be asked to collectively interpret the clusters and their content using a structured approach. The participants will be asked to name each cluster based on their understanding (e.g., the relevance of the JC intervention for organizational aims; supportive and engaged leadership; support for employee strengths use and development). In order to ensure all workshop participants are able to contribute to the collective interpretation of the results, the participants will be divided into small discussion groups during the workshop. Each discussion group will discuss the findings and take notes. After the group discussion, the groups will present their interpretation of the results to the whole group for further discussion. The notes will be collected by the research team After the workshop, the research team will revise the names given to the clusters. The research team will follow guidelines for cluster names when deciding on the final cluster names.

## 3. Discussion

The literature on JC has developed quickly in recent years, but the knowledge about implementing JC interventions that have lasting effects remains vague. If JC interventions are to become an effective strategy to increase employee performance and well-being, then delineating context and process factors that affect their implementation, is a crucial next step in this domain of research. While the research about JC interventions describes what happens during the intervention (e.g., intervention stages, examples of exercises performed during the sessions), the information about the changes that are or should be introduced at the group or organizational level is never or rarely described. From the current literature, it is not clear to what extent and how the environment is prepared to allow for more job crafting and whether this process occurs in parallel to the intervention stages. Therefore, the goal of our study is to understand how these interventions should be designed and implemented to embrace the multilevel perspective to a higher extent. Our contribution is to target the move from the micro-level (individual) to more meso- (group) and macro-levels (organizational).

Involving researchers who study JC and practitioners who work with organizational interventions, will likely lead to an accurate and multifaceted solution, which is closer to actual organizational practice. The resulting maps of the relevant individual, context, intervention, and implementation factors to consider, will become a basis for a practitioners-oriented toolkit that will outline evidence-based recommendations concerning designing and implementing, as well as evaluating JC interventions. The go-zone plot is considered especially useful as a starting point for a deeper analysis of the content of map results. This way, a map can be translated into actionable activities: an “idea-bank” will be developed that will aim to provide decision-makers with a few bullet points describing what they could do. With JC interventions becoming popular among researchers as well as practitioners, such tool-kit can synthesize the most needed knowledge to make the best use of their potential by these groups. Synthesizing this knowledge will also enrich to the current literature concerning the effectiveness of JC interventions.

The guidelines that will be developed via this study could be further investigated in quantitative research (experiments and quasi-experiments, randomized control trials) to test their effectiveness. Distinct combinations of factors important for the sustainability of JC intervention effects could also be analyzed using coincidence analysis (CNA), which has been designed explicitly to answer research questions about combinations of conditions that are minimally necessary or sufficient to produce an outcome, and to identify multiple causal paths to an outcome [43].

Inviting international experts from both practice and academia strengthens the generalizability of the findings. The iterative process, which involves participants in data collection, analysis, and the interpretation of results, further strengthens the validity of the study. Engaging practitioners will also establish ownership and increase the chance for utilization of the knowledge produced by this project. Those who invest time and intellectual effort in the stages of GCM may become more interested and committed to introducing JC interventions into their own workplace. Hence, active involvement with practitioners will also promote the dissemination and application of the results.

Although the format of the data collected in the brainstorming activity might lead to the data being simplified compared to data collected through, e.g., interviews, the format allows for a structured conceptualization process and implicit knowledge to be uncovered [44].

One limitation of using GCM is the self-reported nature of the data. For instance, it has been suggested that the identification of perceived barriers to implementation are often part of a “sense-making” strategy that may have varied meaning in different organizational contexts and may not be directly related to the actual practice [45,46].

## Figures and Tables

**Figure 1 ijerph-19-13922-f001:**
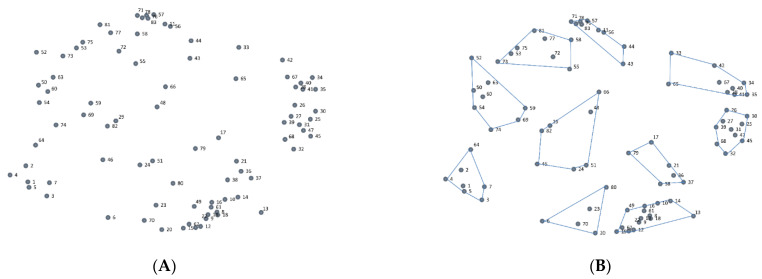
An example of a point map (**A**) and a possible cluster solution (**B**). Note. Dots represent individual statements that have been sorted by the participants.

## Data Availability

The data generated in this project will be available in OSF project: https://osf.io/942mz (accessed on 17 May 2022).

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
