# Peer review of "How Should Job Crafting Interventions Be Implemented to Make Their Effects Last? Protocol for a Group Concept Mapping Study"

_ijerph, 2022, doi:10.3390/ijerph192113922_

Round 1

Reviewer 1 Report

Thank you for the opportunity to review this study protocol. The manuscript presents a novel study design moving towards a better understanding of what factors promote long-lasting effects of job crafting interventions with the use of group concept mapping. Despite finding the protocol overall interesting and useful for research and practice purposes, I have also a few suggestions for improvement, listed below:

1. Since the group concept mapping methodology is key to the study, I found that presenting some additional and more general description of the method would have been useful. For instance, what kind of theoretical or underlying underpinnings does the method have? How has it been used before in work/organizational research? Why is this method useful specifically in the context of job crafting? Moreover, providing some of the points presented in the second paragraph of discussion (lines 266-277 in the manuscript version I received) already in the introduction or methods section could make the transition from the job crafting topics to the more detailed group concept mapping methodology easier to follow.

2. Related to the first point, is there something missing in the second part of the title "Study protocol of group concept mapping"? As far as I understood correctly, group concept mapping is only the general method used in the study to uncover insights related to job crafting intervention effects? As such, the sentence seems somewhat incomplete.

3. For clarity, it may be beneficial to mention the exploratory nature of the study (line 138) also in the abstract.

4. Please add a short summary of research findings concerning avoidance job crafting, as the empirical results seem to only focus on approach crafting (line 78-89) despite both concepts being presented earlier on in the introduction.

5.  How are the potential researcher experts identified in the study (lines 153-154)? What kind of search criteria will be used to get a selection of the relevant articles focusing on job crafting interventions? Will you send an invite to all authors included, or to a subset of authors (e.g., first author)? Are unpublished job crafting intervention studies included in this selection?

6. A more practical question, I was wondering how is "collectively interpreting the clusters and their content" (lines 246-251) facilitated by the research team in the digital workshops? Given that the aim is to include 40 researchers and 40 practitioners, it could be important to make sure that all participants get a chance to voice their opinions, especially in a situation where some participants may have a stronger content knowledge and experience than others.

Minor points:

a. It may be useful to add something like "reaching perceived gains"/ "avoiding perceived negative end-states" on line 68-69 to be more accurate.

b. I was wondering about the sentence "The other is a top-down perspective: the modes by which organizations can prepare the environment to enable employees to craft their jobs after “they are back to work” from the intervention" (lines 133-135) - As interventions should probably induce changes in crafting already during the intervention, is it not important to also support participants at the organizational level already as they go through the intervention procedure?

Author Response

Thank you for the time and your insightful comments about our work. Please see attached our point-by-point response to the comments.

Reviewer 2 Report

Dear Authors,

This is a well-written and interesting paper that describes important research on the question of how to implement JC interventions to make their effects lasting.

Your introduction is very well written and describes your planned study's theoretical and empirical background, which nicely introduces your important three research questions. 

However, I found your methods and the analytical strategy section very detailed and technical, and I wonder if your paper needs all of this detailed and technical information.

I would suggest shortening these sections and making them more informative for those unfamiliar with GCM. For instance, I'm unsure if readers need lines 212-251 to understand your study protocol. I'm also unsure if readers need Figure 1 to understand your analytics. Also, I'm not sure what information one can get from your "Additional file 1" (what, for instance, does the column "page reported" mean? How is this additional file one connected to your text?). Instead, can you give more examples for each step of your GDA for those unfamiliar with this method? 

I would be particularly interested in issues like (1) What do you ask your participants?  What is your instruction? (2) Moreover, do you have Hypotheses? While your research is explorative, you mention that you see "top-down" and "bottom-up" issues related to the question of implementation issues of JC interventions. Do you have some examples of what you expect your participants to mention? (3) You mention "guidelines" in the discussion you may develop based on your findings. You may already discuss these guidelines in your paper's "expected results" part. What my this look like? (4) Can you give examples of each step of your GDA for those unfamiliar with this method? You describe in your discussion the "iterative process" (lines 269 ff). This may be a good outline for your study description. You can describe how you involve participants in a) data collection, b) data analysis, and c) interpretation of results (what are the instructions for each step? What are the expected outcomes? How will you document these outcomes? How does, overall, lead this process uncover implicit knowledge?

Consequently, to me, you can include some parts of your discussion section in your method section. Your discussion section focuses on your methods, but what about the content? The findings? Possible limitations for interpretation of findings? 

Minors

a) Do participants sort every statement or only own statements (line 184)

b) The references may need some further editions (e.g., No. 4, where is it published? Some journal articles are cited by APA standards (e.g., No. 6), others not (e.g., No 7).

c) line 159, this link is not working, the server send a warning for visiting this web-site

I hope these comments may help further improve your paper. I wish you all the best with the study you plan, and I look forward to reading about its results.

Author Response

Thank you for your time and insightful comments about or work. Please, find attached our point-by-point response to the comments.

Round 2

Reviewer 1 Report

Thank you for the careful revision. I have no further comments related to the content. Some text editing may be needed to correct a few minor issues in the revised text (e.g., typos, extra/missing spaces).